# The Effect of *Opuntia ficus* Mucilage Pectin and *Citrus aurantium* Extract Added to a Food Matrix on the Gut Microbiota of Lean Humans and Humans with Obesity

**DOI:** 10.3390/foods13040587

**Published:** 2024-02-15

**Authors:** Nancy Abril Estrada-Sierra, Marisela Gonzalez-Avila, Judith-Esmeralda Urias-Silvas, Gabriel Rincon-Enriquez, Maria Dolores Garcia-Parra, Socorro Josefina Villanueva-Rodriguez

**Affiliations:** Centro de Investigación y Asistencia en Tecnología y Diseño del Estado de Jalisco A.C (CIATEJ), Guadalajara 44270, Mexico; naestrada_al@ciatej.edu.mx (N.A.E.-S.); mgavila@ciatej.mx (M.G.-A.); jurias@ciatej.mx (J.-E.U.-S.); grincon@ciatej.mx (G.R.-E.); dgarcia@ciatej.mx (M.D.G.-P.)

**Keywords:** lean, obese, intestinal microbiota, *C. aurantium*, *Opuntia* cladodes mucilage, food interactions

## Abstract

Experimental studies have provided evidence that physicochemical interactions in the food matrix can modify the biologically beneficial effects of bioactive compounds, including their effect on gut microbiota. This work aimed to evaluate the effect of a food gel matrix with *Opuntia ficus* cladodes mucilage pectin and *Citrus Aurantium* extract on the growth of four beneficial gut bacteria obtained from the fecal microbiota of people who are lean or who have obesity after digestion in the upper digestive system. To accomplish this, a base formulation of *Opuntia ficus* cladodes mucilage with or without *C. aurantium* extract was submitted to an ex vivo fecal fermentation in an automatic and robotic intestinal system. The changes in the intestinal microbiota were determined by means of plate culture and 16S sequencing, while short-chain fatty acids (SCFA) produced in the colon were determined via gas chromatography. In the presence of the extract in formulation, greater growth of *Bifidobacterium* spp. (+1.6 Log_10_ Colonic Forming Unit, UFC) and *Lactobacillus* spp. (+2 Log_10_ UFC) in the microbiota of lean people was observed. Only the growth in *Salmonella* spp. (−1 Log_10_ UFC) from both microbiota was affected in the presence of the extract, which decreased in the ascending colon. SCFA was mainly produced by the microbiota of people who were lean rather than those who had obesity in the presence of the extract, particularly in the ascending colon. The effect of sour orange extract seems to depend on the origin of the microbiota, whether in people who have obesity (25 mM/L) or are lean (39 mM/L).

## 1. Introduction

The intestinal microbiome is made up of a large number and diversity of bacteria, viruses, parasites, and fungi that differ from one person to another [1]. It has been demonstrated that the profile of microorganisms in the intestine is very important for the host’s health since it deals with different vital functions, like the modulation of the immune system, digestion, and absorption of nutrients, as well as defense mechanisms against pathogenic agents [2]. The changes in the microbiota profiles due to external factors such as unbalanced diets and deficient intake of fruits and vegetables, among others, can be negative and a determinant in the development of metabolic diseases [3,4]. It has been shown that the microbiota of people with obesity differs from those who are lean [5]. These changes, both in diversity and composition, have an impact on some biochemical mechanisms; for example, they alter caloric expenditure and energetic homeostasis, and it has also been proven that a high-fat diet decreases the abundance of health-promoting bacteria, such as *Bifidobacterium* spp. and *Lactobacillus* spp., generating an imbalance (dysbiosis) in the microbiota [3,6].

Therefore, some studies have been carried out to understand the mechanisms and factors that determine the balance of intestinal microbiota, as well as the impact of different foods and ingredients on the growth of beneficial bacteria that might have a prebiotic effect [3,7]. Besides its effect on the intestine, probiotics also affect distal organs due to the production of short-chain fatty acid (SCFA) because of bacterial fermentation. Acetic, propionic, and butyric acids are capable of moving through the enterocytes and reaching the blood circulation, so they carry out other functions like decreasing lipolysis and total cholesterol, increasing the absorption of calcium, regulating hormonal production, increasing the general cognitive process, and decreasing the dementia process, among other functions [7].

The main sources of prebiotics are foods high in complex carbohydrates such as pectin, nondigestible polysaccharides like agave fructans, and some secondary metabolites of fruits and vegetables, like flavonoids [8,9]. The *Opuntia ficus indica* cladodes, besides being a plant endemic to Mexico, are widely consumed by its population, and they are rich in nondigestible polysaccharides [10]. When processing the cladode, the vegetal tissue secretes a viscous liquid known as mucilage, rich in structures like pectin. In general, the *Opuntia ficus indica* cladodes mucilage compounds are 25–42% L arabinose, 13% L-rhamnose, 21–40% D-galactose, 22% D-Xylose, and 13% D-galacturonic acid; they have a main chain of acid galacturonic with β-D(1–4) bonds and a rhamnose side chain, and they also contain some peptides, calcium, and some polyphenols like pisidic acid, eucomic acid, and 2-hydroxy-4-(4′hydroxyphenil)butanoic acid [11].

The *Opuntia ficus indica* cladodes’ mucilage has been demonstrated to have biological effects such as hypoglycemic, antioxidant, and anti-inflammatory while also stimulating the growth of probiotic bacteria in people and rats with obesity [12,13,14,15,16]. However, in the literature review, no paper was found that explores the impact of the presence of mucilage as a part of a complex matrix such as food, given the physicochemical interactions, synergies, and antagonisms between functional groups and the impact of this complexity on its prebiotic capacity or that of other bioactive molecules that coexist in the food in which mucilage is incorporated.

Other compounds that have been demonstrated to affect the microbiota are polyphenols. Extracts high in flavonoids from cranberries, green tea, black tea, and grapes have a prebiotic effect [17]. Citrus fruits are a great source of flavonoids. The type and concentration of these compounds change depending on the variety of citrus fruit consumed. The administration of orange juice has been shown to have a prebiotic effect demonstrated with changes in *Lactobacillus* spp., *Bifidobacterium* spp., *Clostridium* spp., and *Enterococcus* spp. [18]. *C. aurantium* is a citrus fruit not frequently consumed in a daily diet; however, it has been widely used by Chinese traditional medicine, mainly in Asian countries, demonstrating different biological effects such as weight control and hypoglycemic and hypolipidemic effects [19,20]. These oranges are rich in sesquiterpenes, monoterpenes, and other compounds such as limonoids, alkaloids, flavonoids, and phenolic acids [21,22]. However, in the literature review, no paper was found that explores the prebiotic effect of this citrus fruit or one of its extracts; neither, as in the case of mucilage, have there been studies that explored the changes in biological effects due to its inclusion in a complex matrix and the given physicochemical interaction that could change the behavior of food.

There are some studies on the impact that the formulation, the elaboration process, and some steps of the digestion process can have on the bioaccessibility of bioactive compounds; all of them discuss the fact that the interactions of bioactive compounds with the ingredients of the food matrix could change the attributed biological effect [23,24,25,26]. For the above, it is hypothesized that the active ingredients of *Citrus aurantium* and *Opuntia* cladode mucilage that reach the colon after the first stage of digestion could have an impact on the growth of some bacteria in the intestinal microbiota. Several studies have been carried out to understand the probiotic and prebiotic phenomenon, using special devices for study [27]. In this study, an automatic and robotic intestinal system (ARIS) was used. The food is submitted to a complete simulation of the digestion process, which simulates the mouth, stomach, and small intestine process controlling the pH and adding enzymes, and at the end, the previously digested food consecutively passes through three reactors that simulate the ascending, transverse, and descending colon. The colonic segment of the digestion process may be inoculated with fecal samples from humans with different physiological and dietary conditions based on the phenomenon to be studied [28]; a schematic figure of the equipment is presented in Figure 1; this provides an idea of the effect of food on the microbiota of different population groups, for example, imbalanced microbiota produced by pathologies such as obesity, and balanced microbiota present in people without overweight or obesity. The objective of this study was to evaluate the effect of the administration of a food gel matrix, whose main ingredients were *Opuntia* cladodes mucilage and a *C. aurantium* extract, on the growth of beneficial and potentially pathogenic bacteria families present in microbiota from people with obesity and lean people.

## 2. Materials and Methods

### 2.1. Food Matrix

A semi-solid food matrix made of gelatin (9%) and the residue of the osmotic dehydration of *Opuntia* cladodes (high in mucilage and sucrose, 60° brix, 83%) added with aqueous extract from *C. aurantium* was designed, called extract matrix (EM). The *C. aurantium* extract was obtained from fresh residue of a green orange (all except juice) harvested in the Summer of 2018 collected from trees from streets in Guadalajara, made with a ratio of 1:3 *w*/*v* of solvent (8% of the formulation), The presence of p-synephrine (14 mg/g of extract), naringin (33 mg/g of extract), and neohesperidin (29 mg/g of extract) in the orange extract was verified by means of high-resolution liquid chromatography (HPLC) with UV detector [22]. All details of the election of extracts are reported in previous research cited in [22]. On the other hand, a matrix control was prepared with a solution of ascorbic acid made in relation 0.3:1 (*w*/*v*) to replace the 8% of the *C. aurantium* extract with the goal of ensuring the same formulation and pH (4.3) with no flavonoids in it, was called the no extract matrix (NEM).

### 2.2. Gastrointestinal Conditions and the Obtention of Microbiota

The ARIS was used to simulate the digestive process (Figure 1); it simulates peristaltic movements and gastric emptying, in addition to allowing inoculation with human microbiota in the last portions of the digestive tract (patented devise WO2017/065595 A2 [29]). The simulator has five double-wall glass reactors (1000 mL) sequentially connected, simulating the stomach, the small intestine, and the ascending (AC), transverse (TC), and descending (DC) sections of the large intestine; these last three sections were inoculated with the microbiota of people with obesity or lean people.

To inoculate the portions of the large intestine, fecal samples obtained from 32 volunteers, 13 men and 19 women within an age range of 22–64 years old, were processed. Despite not carrying out a direct clinical intervention, the samples were obtained in accord with the accepted protocol by the ethics and research hospital committee of the public health hospital, Institute of Security and Social Services of State Workers, with the number ISSSTE/CEI/2018/241. To not change the original microbiota, none of the participants had a background of antibiotic treatment 6 months before the study, nor at the moment of the obtention of the sample. The participants were asked to avoid consuming fermented foods one week before the obtention of the sample. The 32 participants were classified according to their body mass index, forming two groups: a group of 16 volunteers with normal weight (IMC 22.2 ± 1.2) and another group with 16 volunteers with obesity (IMC 32.6 ± 2.9).

To obtain the microbial sample, 1 g fecal material from each volunteer was taken and diluted with 10 mL of phosphate buffer previously sterilized (0.1 mol/L, pH 7). Each sample was homogenized and centrifuged at 5000 rpm for 10 min at 4 °C.

The microbiota mix from individuals who are lean (LN) or individuals with obesity (OB) was obtained by mixing 3 mL of supernatant from each one of 16 samples corresponding to each group and inoculated in each reactor of the ARIS system, which represents the three segments of the colon (AC, TC, and DC); each one of these reactors contained 100 mL of a stabilizing media made from yeast extract and casein peptone among other ingredients [28]. In parallel, the supernatant was microbiologically analyzed to determine and compare the initial microbiota of the two study groups.

The already inoculated reactors were kept at 37 °C in constant agitation (120 rpm), adjusting the pH to the established conditions for each portion of the colon (AC = pH 5.5–6, TC = pH 6–6.5, and DC = 6.5–7), with a solution of NaOH (3 M) or HCl (0.5 M) as needed. These were allowed to stabilize for one week, adding daily 10 mL of a multivitamin solution (0.5 g/L, viterra^®^ plus by Pfizer, Monterrey, Mexico) and adjusting the pH according to the conditions of each portion of the colon to allow the microbiota to remain balanced.

### 2.3. Administration of the Food Matrix to the Digestive Tract Simulator (ARIS)

After one week of the initial stabilization in the reactors of the large intestine, the digestion process started with the samples of interest. The experiment was carried out according to the report by Garcia-Gamboa et al. [28].

Simulation of digestion in the mouth: Eight grams of the matrix was mixed with 8 mL of artificial saliva, formulated as 0.8 mL of α-amylase (Sigma-Aldrich, St. Louis, MO, USA, cat. G-4862) at 1500 U and 5.6 mL from a mix of KCl (J.T Baker, Phillipsburg, NJ, USA, cat. 3040) at 15.1 mM, KH_2_PO_4_ (Sigma-Aldrich, cat. P-5655) at 3.7 mM, NaHCO_3_ (Sigma-Aldrich, cat. 900506) at 13.6 mM, MgCl_2_(H_2_O)_6_ (Sigma-Aldrich, cat. 7791-18-6) at 0.15 Mm, 0.04 mL of CaCl_2_ (Sigma-Aldrich, cat. 449709) at 0.3 M, and 1.56 mL distilled water. Once the mixture was made, the pH was adjusted to 6.5–6.8, and it remained in agitation at a constant temperature (120 rpm and 37 °C) for 2 min.

Simulation of the stomach: The simulated bolus from the mouth was mixed with 100 mL of a Western diet separately for the healthy donor (1500 kcal, 65% carbohydrates, 15% proteins, and 20% lipids) or donor with obesity (2500 kcal, 50% carbohydrates, 20% proteins, and 30% lipids), the pH was adjusted to 2.5, 0.125 g of pepsin (Hycel, cat. 155) (10,000 U/Kg) was added, and the mixture was kept under constant temperature and agitation (120 rpm, and 37 °C) for 2 h.

Simulation of the small intestine: The total volume of digested product in the stomach was mixed with 10 mL of pancreatic enzymes solution (25,000 Ug/L lipase, 100,000 Ug/L amylase, and 60,000 Ug/L protease, Sigma-Aldrich, cat. P1625), the pH was adjusted to 5, and the mixture was kept under constant agitation and temperature (120 rpm and 37 °C) for 4 h.

For the colon simulation, each reactor (ascending, transverse, and descending sections) was inoculated with intestinal microbiota from the lean population or the population with obesity in separate studies.

Simulation of the ascending colon (AC): The total of the digested product formed in the small intestine (100 mL) was added to the inoculated reactor, the pH was adjusted to 5.5, and the mixture was kept under constant agitation and temperature (120 rpm and 37 °C) for 18 h. 

Simulation of the transverse colon (TC): 100 mL of the digested product from AC was added to the TC reactor, the pH was adjusted to 6.5, and the mixture was kept under constant agitation and temperature (120 rpm and 37 °C) for 32 h.

Simulation of the descending colon (DC): 100 hundred mL of the digested product from TC was added to DC, the pH was adjusted to 6, and the mixture was kept under constant agitation and temperature (120 rpm and 37 °C) for 16 h. 

The completed digestive process was cyclic, so it was carried out consecutively until completing 3 cycles of the digestive process. Each cycle lasted 72 h. The first samples were taken on day 0 (control or baseline period) and then on the third day of the process, which was considered as cycle 1, the sixth day as cycle 2 (D2), and the ninth day as cycle 3 (D3). This cyclic process was applied to evaluate the prebiotic effect of the two different matrices under study (matrix with extract and matrix without extract) on the two types of microbiota (lean and obese), resulting in four different simulation processes.

### 2.4. Determination of Antioxidant Activity and Total Flavonoid Content in the Mouth, Stomach, and Small Intestine

To understand the liberation of food components during the first steps of the simulated digestion, the changes in antioxidant capacity were determined according to Nakajima et al. [30] with modifications in the sample volume, where 50 µL of each digestion sample and 150 µL of the DPPH (1 M) solution were mixed and added in a microplate to stand in darkness for 30 min; then, the absorbance was read at 520 nm. The results were expressed in equivalent micromoles of Trolox (µMET/L) using a calibration curve from 0 to 1250 µMET/L with increases of 100 µMET/L (R^2^ = 0.991).

The change in total flavonoid content (TFC) was determined according to Tounsi et al. [31], where 100 µL of each digestion sample was mixed with 250 µL distilled water and 75 μL of freshly prepared 5% NaNO_2_ (Sigma-Aldrich, St. Louis, MO, USA) solution and incubated at room temperature for 6 min, followed by the addition of 150 μL of 10% AlCl_3_ (Sigma-Aldrich, St. Louis, MO, USA) solution for 5 min. Next, 500 μL of 1 M NaOH plus 425 μL distilled water was added, shaken vigorously, and allowed to incubate for 30 min. Thereafter, the absorbance at 510 nm was measured using a Multiskan GO spectrophotometer (Thermo Scientific^TM^, Vantao, Finland). The total flavonoid content was calculated as mg of quercetin equivalent (QE)/g waste. The calibration curve range was 0–800 mg/L with increases of 100 mg/L (R^2^ = 0.99).

### 2.5. Microbiological Analysis during Large Intestine Simulation

The initial fecal samples of each subject, as well as those taken during the digestive process in the ARIS (0, 3, 6, and 9 days), were treated and analyzed in the same way: 1 mL of the samples was dissolved in peptone water to prepare serial dilutions, the cultivation of bacteria groups was made by using MRS (Man, Rogosa and Sharpe agar) for *Lactobacillus* spp., LIA (Lysine Iron Agar, BD Bioxon™ Ciudad de Mexico, Mexico) for *Salmonella* spp. (24 h, microaerophile), BSM (Bifidus Selective Medium-agar, Sigma-Aldrich, St. Louis, MO, USA) for *Bifidobacterium* spp. (48 h, anaerobic), and TSC (Tryptose Sulfite Cycloserine, BD Bioxon™ Ciudad de Mexico, Mexico) for *Clostridium* spp. (24 h, anaerobic), and the samples were incubated at 37 °C for 24 to 48 h [28].

The bacterial growth is presented in a logarithmic scale, subtracting the control using the following equation:Growth = Ln(N/N_0_),
where N represents the bacterial count at a certain sampling point, and N_0_ represents the initial bacterial count. All the measurements were made in triplicate, and the data were represented by the average and standard deviation.

### 2.6. Analysis of Short-Chain Fatty Acids during the Large Intestine Simulation

Short-chain fatty acids (SCFA) were identified and quantified via gas chromatography with a flame ionization detector in a GC 2010 chromatograph of the brand Shimatzu^®^ (Toyko, Japan), using the method reported by Pietro-Femia et al. [32] with some modifications. To 250 µL of the sample from the colon digestion process, 25 µL of phosphoric acid solution (5 M) and 250 µL diethyl ether were added. The mixture was homogenized and subsequently centrifuged at 13,000 rpm for 3 min (analyte recovery average of 80%). The ether phase was injected directly onto a column HP-FFAPP (30 m × 0.250 mm × 0.25 µm, columns Agilent JW GC) at 180 °C, using N_2_ as a carrier gas, and the flow was 1 mL min; the run time for each analysis was 15 min, and the temperature of the detector was fixed at 230 °C. Quantification was carried out using calibration curves of acetic (R^2^ = 0.9979), propionic (R^2^ = 0.9993), and butyric (R^2^ = 0.9975) diluted in water to a concentration range of 40, 20, 15, 10, 5, 2.5, 1.25, and 0.75 µM/L [32].

### 2.7. Taxonomic Analysis of Intestinal Microbiota via Bacterial DNA Extraction, 16S RNA Amplification, Library Construction, and Massive Sequencing

The isolation and concentration of the genetic material were carried out with a sample obtained from each of the different reactors of the large intestine (AC, TC, and DC) once the digestive process was completed (third cycle “D3”), analyzing a total of 12 samples. For each one of the samples (20 mL), solid residues were removed by centrifugation (2000 rpm, 5 min). Fifteen mL of the supernatant was centrifuged at 8000 rpm for 8 min at 25 °C to precipitate the genetic material. The precipitate was subjected to a washing process, which consists of mixing an equal part of the precipitate (approximately 1 g, depending on the sampling process) and recently prepared PBS solution (phosphate-buffered saline solution) (*w*/*v*), which was subsequently homogenized for 1 min using a vortex and centrifuged at 8000 rpm for 5 min at 4 °C; finally, the supernatant was discarded. This washing process was repeated 4 more times. Then, mechanical hydrolysis was carried out by adding 50 µL of PBS and heating at 95 °C for 3 min, using the DNA extraction kit DNeasy afterward by Qiagen^®^ following the instructions of the manufacturer to carry out such extraction.

For the amplification of the 16S rRNA gene, specific primers were used (direct primer 5′-TCGTCGGCAGCGTCAGATGTGTATAAGAGACAGCCTACGGGNGGCWGCAG-3′ and reverse primer 5′-GTCTCGTGGGCTCGGAGATGTGTATAAGAGACGGATACHV3′/VGTGTGTGGGGCTCGGAGATGTGTATAAGAGACGGATACHV3 that covers the hypervariable regions V3/V4 of the ARNr 16S gene.

The library construction was carried out according to the Illumina protocol of metagenomic sequencing library preparation 16S for the amplification, purification, and index union. The amplicons were quantified using a Qubit^®^ 3.0 fluorometer (Thermo Fisher Scientific, Waltham, MA, USA). Equimolar amounts of DNA were combined and sequenced per sample in the facilities of the Sequencing Service of FISABIO (Valencia, Spain) using the reagent kit MiSeq^®^ version 3 (Illumina, San Diego, CA, USA) in a desk sequencer MiSeq (2 × 300 bp paired-end readings) (Illumina). The internal controls of extraction and amplification were also used with the samples.

### 2.8. Statistical Analysis

All the experiments were made in triplicate to evaluate the effect of the microbiota origin: obese or lean; a comparison of means was carried out by using Student’s T test to evaluate the effect of time of digestion and the presence of extract on the microbiota, a generalized linear model was applied (GLM). The significance of the difference between means was determined by Tukey’s multiple range test considering a *p* < 0.05. All the statistical analyses were carried out using the software Statgraphics Centurion XV (Manugistics Inc., Statistical Graphics Corporation, 1993, Rockville, MA, USA).

## 3. Results and Discussion

### 3.1. Changes in Antioxidant Activity and Total Flavonoid Content during the First Part of the Digestion Process

The change in antioxidant activity and total flavonoid content of EM and NEM during the first part of digestion are present in Figure 2. When the EM was administered, the antioxidant activity was higher in the small intestine than in the other parts of the simulation (Figure 2a), while in total flavonoid content, there were no significant differences between the stomach and the small intestine (Figure 2b). This behavior was similar to that reported by other authors who submitted some rich antioxidant food via the digestion process, such as carrot powder [33] in sea buckthorn berry juice [34] or in quinoa seeds, flakes, and sprouts [35]. This behavior may be due to the breakdown of physicochemical interactions between the bioactive compounds and the ingredients of the matrix that release the bioactive compounds throughout the digestion process [36].

Once the liberation of bioactive compounds is achieved, it is important to explore the effect of these free bioactive compounds on the growth of bacteria from different microbiota in the large intestine. For this, the digestion process was carried out simulating the portions of the large intestine where different types of microbiotas were included. Studying two conditions in which the large intestine is inoculated with microbiota from participants with obesity or microbiota from lean participants.

### 3.2. Abundance and Identity of Initial Microbiota: Comparison between Microbiota from People Who Are Lean and People with Obesity

The four analyzed bacterial genera, before adding any matrix to the reactors, were more abundant in the microbiota of the those who have obesity compared to the microbiota of the lean (Figure 3). This difference was significant only in the case of *Lactobacillus* spp. (Figure 3a, *p* = 0.010) and *Clostridium* spp. (Figure 3c, *p* = 0.038). The presence of some species of *Lactobacillus* spp. has been associated with weight gain [37,38], given that they can easily degrade polysaccharides, turning them into SCFA, which are used by the host as an energy source, increasing the amount of stored energy in people with obesity (15 to 20% more) compared to the amount of stored energy in a lean person who consumes the same food portions [39]. Several authors have found the same significant difference [5,40], and some others observed a higher diversity of species of the same genus in the microbiota of people with obesity than in the microbiota of lean people, which could be associated with a gut dysbiosis that could alter the energy absorption [41].

### 3.3. Taxonomic Analysis

After three cycles of digestive processes, samples were taken for taxonomic analysis (Figure 4a–c), and it was found that the most abundant were *Firmicutes* in a proportion between 79 and 99%, followed by *Proteobacteria* in a proportion of 10 to 0.4%, depending on the sample analyzed. This behavior is not consistent with some reports in the bibliography, where it was shown that in healthy adults, the proportion of *Firmicutes* was about 50%, with *Bacteroidetes* (48%) in practically the same proportion; this could be due to interindividual differences in microbiota diversity among the 16 subjects participating in each of the two groups [2]. Andoh et al. [42] reported a higher proportion of *Firmicutes* and attributed this dissimilarity with other reports on the same topic to differences in the diet, habits, hygiene, and even the genetic aspect of the Japanese population compared to the Western population, which is the most frequent object of study published in the scientific literature. Likewise, the differences in this project could be because the population is Latin. On the other hand, it could also impact some operational conditions of ARIS.

In a general overview, 67 bacterial species were found that belonged to 34 families (Table 1); 44 of them were common in both groups (66%), and 34% appeared only in one type of microbiota, with a higher diversity of the microbiota in participants with obesity with 15 differentiators species (22%), compared to the microbiota in lean participants with 8 differentiators species (6%) (Figure 4d). This is consistent with what was found in the microbiological analysis of the initial samples (Figure 2) and the report in the bibliography. (Table 1, Figure 4), which may impact the microbiota’s metabolic behavior when administering the different matrices.

In AC, it was found that in the presence of NEM, the microbiota of the OB group showed a higher amount of *Firmicutes* than the microbiota of the LN group (Figure 4a). Some research reported a higher proportion of firmicutes in OB than in LN people [37,43]. This behavior was not observed in the other segments of the colon (Figure 4b,c), and it could be because there are ideal conditions for growth in the AC, like the pH and the presence of abundant carbohydrates as a source of energy, so the “normal” behavior of bacteria can be observed [31,36]. Specifically, in the EM OB group there was observed an increase in the abundance of proteobacteria and a decrease in the proportion of *Firmicutes* regardless of the simulated section of the colon (Figure 4a–c). In contrast, the microbiota of the LN group was not affected by the presence of the extract. Similar behavior was observed by Lee et al. [44] when blueberry was administered to high-fat-diet mice and its referrer; this could be partly due to the presence of flavonoids that enhance the presence of some bacteria such as Fusobacterium, which is present in our taxonomic analysis (Table 1). In their case, these changes improved the integrity of the gut barrier. This effect of the extract will be considered evidence of the impact of bioactive compounds on the colon’s microorganisms.

Table 1 shows the influence of the presence or not of the extract in the matrix and the origin of the microbiota on the type of families present in each section of the colon. First, it was found that the presence of the *Lacnospiraceae* family is not affected by any of the study variables since it is present in both microbiotas, in all simulated portions of the colon, and when both matrices were administered.

Five families were present just in the microbiota of participants with obesity, and the study variables, such as the simulated colon segment and the type of matrix administered, did not have a significant effect on them (Table 1). On the other hand, there are 12 families present in the microbiota of participants with obesity that also were affected by the colon segment and the presence of extract. Notably, *Comamonadaceae* was detected when the EM matrix was administered in both AC and TC, and it was consistently present in the DC under both matrices. This bacterial family is recognized as probiotic, with its growth reportedly influenced by the presence of other probiotic families, such as L. Casei [45]. 

Moreover, *Alcaligenaceae* exclusively emerged in the TC and DC upon the administration of EM (Table 1). This bacterial family is associated with inflammation and is commonly identified in the intestinal microbiota of individuals with obesity [46]. Simultaneously, the presence of EM stimulated *Synergistaceae* in the DC. This family plays a crucial role in acetate oxidation, potentially contributing to the production of additional metabolites, including butyrate or valerate, both being short-chain fatty acids (SCFA) present in the digestive process [47].

On the other hand, five bacterial families were identified in the DC across both matrices (Table 1). While information is lacking for some regarding their presence in the intestinal microbiota or associated impacts, others are reported with some functions, such as *Fusobacteriaceae*, known for its association with systemic inflammation and frequent occurrence in obese microbiota [48], and *Arcobacteraceae*, linked to gastrointestinal symptoms like diarrhea, exhibit a diversity that includes nonpathogenic species capable of producing beneficial short-chain fatty acids (SCFA) and possessing a colonizing capacity [49]. Additionally, *Tannerellaceae* has been observed to play a role in breaking down flavonoids and benzene rings, with the growth of these bacterial groups stimulated in the presence of such compounds [50].

In the microbiota of lean participants, five families were exclusively identified, with two of them exhibiting no significant impact from any of the study variables. The remaining three were affected by the variable segment of the colon but not by the presence or absence of extract. *Bifidobacteriaceae* was present just in the CT and CD and is one of the most common probiotic bacteria [51]. While *Coriobacteriaceae* and *Leuconostocaceae* appeared in the CD, both play roles in the efficient utilization of glucose and contribute to its homeostasis [52,53].

Eleven bacterial families inhabit both types of intestinal microbiota. Among them, four seem to exhibit a random response to the administration or not of the extract and the simulation of colon segments, while the remaining seven were influenced by at least one study factor affecting their presence. For instance, the *Ocillospiraceae* family was exclusively present when EM was administered in AC and TC; subsequently, it was found in DC and AC from lean participants regardless of the matrix administered. This bacterial family has been demonstrated to alleviate chronic inflammation and obesity directly or indirectly [54].

*Desulfovibrionaceae* is observed in the DC irrespective of the administered matrix type. Additionally, in the LN microbiota, it was found in the TC upon the administration of EM. This bacterial family stimulated the use of SCFA and is associated with metabolic diseases [55]. *Eubacteriaceae* appeared in the TC and DC in OB microbiota while appearing in all the colon simulations in LN microbiota; it is considered a prebiotic due to the production of SCFA, and it is associated with a decrease in cholesterol [56]. *Anaerovoraceae* and *Atopobiaceae* in OB microbiota were exclusively present in the DC. In contrast, in the LN microbiota, the first bacterial family appeared in both AC and TC, while the second one emerged in DC and AC only when the EM matrix was administered. Both *Anaerovoraceae* and *Atopobiaceae* families are known to produce SCFA, and experience increased growth when study subjects consume fermented foods or in the presence of prebiotic spices [57,58,59].

In contrast to the OB microbiota, the sole pathogenic bacterial family that specifically emerged in the LN microbiota was *Veillonellaceae* in the AC. This family is linked to an increase in inflammatory markers in irritable bowel syndrome [60]. Hence, it is suggested that matrices may, to some extent, prevent the imbalance of the microbiota, a protective effect that seems to be diminished in the microbiota of participants with obesity.

### 3.4. Effect of the Food Matrix on Bacterial Growth

The effect of the study variables (origin of the microbiota, presence of extract, and section of the colon) on the growth of the four species in the experiment did not show a general pattern; this effect depended on the species. In the case of *Lactobacillus* spp. (*p* = 0.001) and *Bifidobacterium* spp. (*p* = 0.003), significant growth was observed as the digestion progressed, regardless of the origin of the microbiota, with the digestive cycle 1(D1) presenting the lowest growth (Figure 5a,b); in addition, *Salmonella* spp. had limited and lower growth than other bacteria, and these variations over time could be due to the characteristics of the group that responded better to certain conditions. This behavior is consistent with what was reported by Gil-Sánchez et al. [61] for beneficial and potentially pathogenic bacteria, respectively. The impact of the rest of the factors depended on the characteristics of each analyzed bacterial group; therefore, the growth detail of each group is analyzed in the following sections.

#### 3.4.1. Effect of the Food Matrix on Probiotic Bacteria Growth

The variables that had a significant effect on the growth of *Lactobacillus* spp. (Figure 6) were the colon section and type of matrix. The highest growth of *Lactobacillus* spp. (Figure 6) was observed in the AC in the LN EM group (1.8 CFU). In the TC (2.2 CFU), the highest growth was observed both in OB and NEM.

In the presence of the extract (EM), the population of *Lactobacillus* spp. from the microbiota of lean participants (LN) showed a decrease as it passed through the three segments of the colon (Figure 6); however, in the absence of extract, the behavior was the opposite, and an increase was observed as these lactobacilli crossed the colon. The growth of some species of *Lactobacillus* decreases in the presence of phenolic acids [62]. Based on the taxonomic analysis, the *Weissella* genus was found exclusively in the microbiota of lean people; this belongs to the order *Lactobacillaes*, and this bacterial genus demonstrated the capacity to form phenolic acids like chlorogenic acid. These polyphenol acids have been shown to have an antibacterial effect due to the fact that they could acidify the medium and limit the growth of some species of *Lactobacillus*, similar to that observed in the EM TC [63]. On the other hand, the polysaccharides from *Opuntia ficus* indica mucilage increased the growth of probiotic bacteria [12]; the NEM, rich in pectins like mucilage and sugars, due to the presence of the osmotic dehydration residue of the *Opuntia* cladode, constitutes an energy source without the inhibitory effect of flavonoids that could explain the highest growth.

In the OB microbiota, the presence of the extract (EM) had a significant effect on *Lactobacillus* spp. growth in each segment. If we compare the NEM and EM (Figure 6), the EM increased the *Lactobacillus* spp. counts in the AC significantly decreased them in the TC and significantly increased them again in the DC. The decrease in the TC may be due to the antagonistic effect exerted by phenolic acids that formed as a result of the breakdown of polyphenols [40], while the growth in the AC could be because the flavonoids lost their glycoside residue in addition to being utilized as an energy source by *Lactobacilli* to help to form phenolic acids. In the DC, it could be due to the elimination of the antagonist effect of phenolic acid due to the interaction with some polysaccharide residues from OCM [63].

For *Bifidobacterium* spp., the segment of the colon studied had no significant effect, while the type of matrix did. The EM had significant (Figure 7) growth in the LN microbiota. Bussolo de Souza et al. [64] found a higher growth of *Bifidobacterium* in lean people than in people with obesity, and they attributed these changes to differences in the microbiota of lean people, while Munekata et al. [65] observed the significant growth of *Bifidobacterium* spp. by administering different extracts of rosemary; probably, the growth of this bacterial species is favored by the presence of flavonoids. In the present study, according to the taxonomic analysis, *Bifidobacterium longum* was found in LN, while in OB, it was not (Figure 7), and it was found in the greatest relative abundance when EM was administered. The presence of different flavonoids in some extracts, mainly aglycones, increased the growth of *Bifidobacterium* spp. when compared to the control of flavonoids. Some microorganisms like *Bifidobacterium adolescentis* could degrade flavonoids like quercetin [65,66].

#### 3.4.2. Effect of the Food Matrix on the Growth of Potentially Pathogenic Microbiota

In *Clostridium* spp., an effect of the presence of extract and the growth time depending on the section was observed (Figure 8). In these species, the ideal is to observe a control in their growth. In the NEM group, the lowest growth of *Clostridium* spp. was observed in the TC on D1, while in the EM group, the lowest growth was in the TC on D3 (Figure 8). This behavior is consistent with the reported growth of *Clostridium leptum*, which decreases over the time of digestion with grape seeds and cranberry extracts [67]. Regardless of the presence (EM) or absence (NEM) of the extract, the highest growth was observed in the AC during D3. This could be related to the type of *Clostridium* spp. in the microbiota. Some research did not observe significant differences in the growth of *Clostridium* spp. when administering the *Opuntia* cladodes mucilage [14]. In contrast, others observed that administrating some extract rich in polyphenols of *Ribes* spp. or an extract of grape seed induced the growth of *Clostridium* cluster XIVa in the first hour of fermentation, which decreased over time [61,68]. The families present in the *Clostridium* cluster XIVa have the capacity to degrade pectins and flavonoids due to the presence of some enzymes like β-glucosidase and pectinmetilesterase dehydrogenase. The taxonomic analysis allowed the identification of some species that belong to the family of *Lachnospiraceas* in these microbiota (Table 1), which are considered part of the *Clostridium* cluster XIVa that might be stimulated by the pectin and flavonoids present in the samples, stimulating at the same time the production of SCFA, which has proved to be important for the overall health of the host [69,70,71].

An effect of the section of the colon, the origin of microbiota, and the type of matrix on the growth of *Salmonella* spp. (Figure 9) was observed (*p* = 0.001). According to the bibliography, a control or decrease in growth should be observed. When EM was administered, the genus *Salmonella* from both OB and LN microbiotas decreased in the AC (Figure 9), while when NEM was administrated, the growth only decreased in the OB microbiota in the DC. This can be due to, on the one hand, the presence of pectin in both matrices; for example, extracts of pectin and its hydrolyzed from mango have an antibacterial effect on some species of pathogenic *Salmonella*, which cause typhoid [72], while on the other hand, specifically in the AC (pH 5.5–6), it could be due to the presence of SCFA from the presence of *Lactobacillus* spp., shown in Figure 6, that could lower the pH (4.9) and generate this antibacterial activity. The different OB and LN microbiota behavior can be due to the difference in the sub-species in each microbiota, such as the presence of bacteria of the genus *Citrobacter* that, according to the taxonomic analysis performed on the research samples (Table 1), are only found in the microbiota of those with obesity and share biochemical and serologic similarities with the *Salmonella* spp. [72,73].

The highest growth was observed when EM was administered in the LN microbiota in the TC and DC. Previous research found that anthocyanin, from blueberry extract, with a chemical structure similar to that of deglycosylated flavonoids, stimulated *Salmonella typhimurium* growth, but after a while, the growth stopped [74]. Others found that an extract rich in aglycone naringenin form naringin inhibited the growth of *S. typhimurium*; this is one flavonoid present in *C. auraintium* extract [75].

### 3.5. Short-Chain Fatty Acid Production

The presence (EM) or absence (NEM) of extract in the matrix, the origin of the microbiota (LN vs. OB), the section of the colon (AC, DC, and TC), and the culture time (cycle) showed a significant effect on the SCFA production.

The presence of extract (EM) increased the production of all SCFA over time (Figure 10); on day nine (D9), a higher concentration was found compared to the control “D0” (*p* = 0.001). This behavior is consistent with the bacterial growth observed, which means that the bacteria from the different segments of the colon used the matrix as a source of energy, generating sub-products of SCFA.

In the total SCFA (Figure 11a), the highest production was in the EM in the DC-LN, followed by the NEM in the DC-OB and TC-LN. The results regarding the stimulation of the production of SCFA compounds are contradictory regarding flavonoids. It has been tested with different ingredients like red wine [76], some standard phenolic acids like caffeic, chlorogenic, and ferulic acid [77], as well as orange juice [18], and all of these investigations found an influence of time on the fermentation that raises the concentration of SCFA, while with Chilean currants [68] and cranberries [67], there was no change demonstrated in the concentration after adding the extract rich in polyphenols. This has to do with factors like the chemical structure of the flavonoid, the pH of the medium, and the type of microbiota in the experiment. Regarding the simulated segments of the colon, the highest concentration was observed in the DC (Figure 11a, *p* = 0.0001). This is consistent with the report by several authors who found a higher concentration of acetic, propionic, and butyric acids in the DC, regardless of the type of matrix administrated [61,77]. According to Tuohy et al. [78], the interaction of polyphenols and fiber generates a delay in the fermentation of fiber, leading to more carbohydrates in the colon. In terms of flavonoids, the formation of SCFA is closely related to their structure; when these are glycosylated, they do not stimulate as much growth compared to when they are in their simplest forms like aglycones, which proves that the highest production of SCFA, after administering the EM, was observed in the DC, since at this point of the digestive process, the original structure of the flavonoid was metabolized [4,64,79].

Considering the proportions (Figure 11b) of SCFA, the presence of the extract had a different impact on the function of the origin of microbiota (LN and OB) and the segment of the colon. It was observed that in the AC, despite the presence (EM) or absence (NEM) of the extract, the proportion of fatty acid did not change; the least produced fatty acid was butyric acid, while acetic and propionic acids had similar proportions (Figure 11b). This is consistent with the literature, where it was found that the highest concentration of SCFA occurred with acetic and propionic acids and in a lower proportion for butyric acid [77,80]. In the TC and DC, these proportions were affected by the presence of the extract in the function of the origin microbiotas. When the extract was administered to the LN microbiota, the proportion of acetic acid decreased, and the propionic acid increased in the DC, while in the OB microbiota, all SCFA had similar proportions. Higher production of propionic acid has been found when administering *Opuntia* cladodes mucilage [14], inulin and fiber [64], some isolated flavonoids [77], or a mixture of fiber and flavonoids [81]. Aguirre et al. [81] also found a higher increase in propionate by administering the mixtures in the microbiota of people with obesity more than lean people. The same behavior was observed by Bussolo de Souza et al. [64]. Sadeghi et al. [76] pointed out that the variations in the concentration of SCFA may be associated with the differences between the compared microbiota. It has been proven that propionate is mainly produced by bacteria *Clostridium cluster IV and XIVa*, while acetic acid is mainly produced by *Bifidobacterium* spp. [80,82]; so, it is believed that the main differences in the concentration of SCFA could be due to the differences observed in the microbiota used in this project.

## 4. Conclusions

Both matrices stimulated the growth of beneficial bacteria and produced SCFA; however, the changes in the microbiota due to the presence of the extract are influenced mostly by the differences in species that form the microbiota of people with obesity and lean people. The sensitivity of these different species might seem determinant partly due to their capacities for metabolizing or degrading the different types of prebiotics—in this case, flavonoids. The same factor seems to induce different evolution patterns as time passes and as they move through the three segments of the colon.

Although certain prebiotic effect trends were observed, a unique pattern was not observed due to the capacity to metabolize the flavonoids of each microbiota tested due to the high gamma of variables, which must be controlled. Different mechanisms will be specifically affected by time, which will feed back to the growing microorganisms, changes in growth cycles depending on the evolution of metabolites, and maybe the dispute between species by those metabolites.

Both matrices stimulate the growth of *Lactobacillus* spp. *and Bifidobacterium* spp. but also control the growth of potentially pathogenic bacteria like *Clostridium* spp. and *Salmonella* spp.; these matrices, under these conditions, could be considered prebiotics.

## Figures and Tables

**Figure 1 foods-13-00587-f001:**
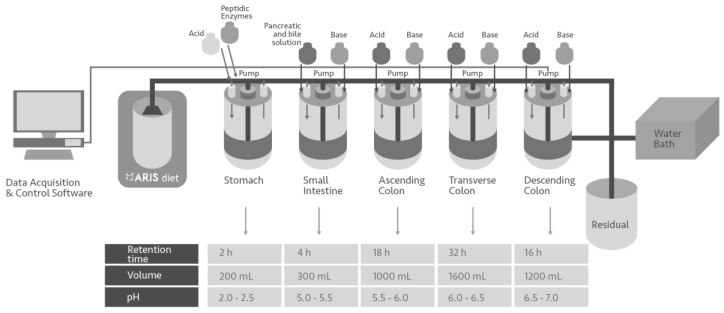
The schematization of the automatic robotic intestinal system (ARIS) [28].

**Figure 2 foods-13-00587-f002:**
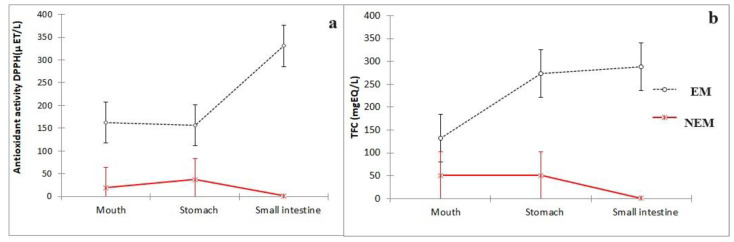
Comparison of (**a**) antioxidant activity and (**b**) total flavonoid content (TFC) in terms of the extract matrix (EM) and no extract matrix (NEM) in the first part of the digestion process. The bar represents the standard deviation of the samples (*n* = 3).

**Figure 3 foods-13-00587-f003:**
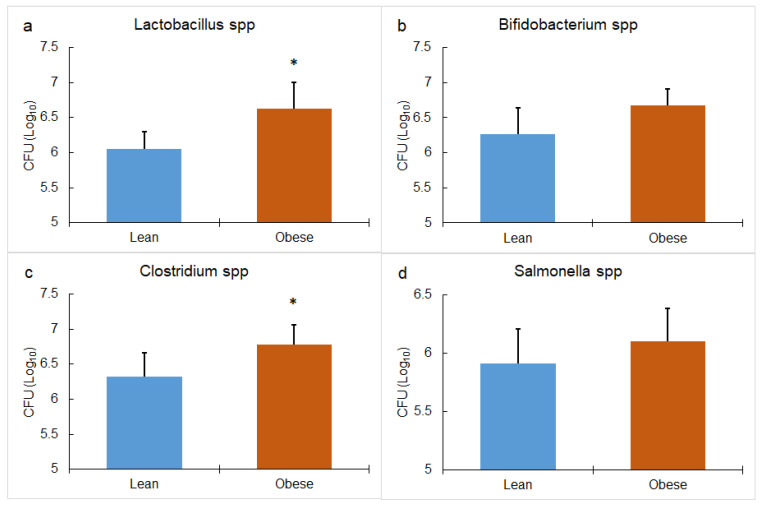
Comparison of the microbiota of people with obesity and lean people in the initial inoculum of (**a**) *Lactobacillus* spp. (**b**) *Bifidobacterium* spp., (**c**) *Clostridium* spp., and (**d**) *Salmonella* spp. * means a significant difference between lean and obese *p* < 0.005 (by Tukey’s multiple range test). The bar represents the standard deviation of the samples (*n* = 3).

**Figure 4 foods-13-00587-f004:**
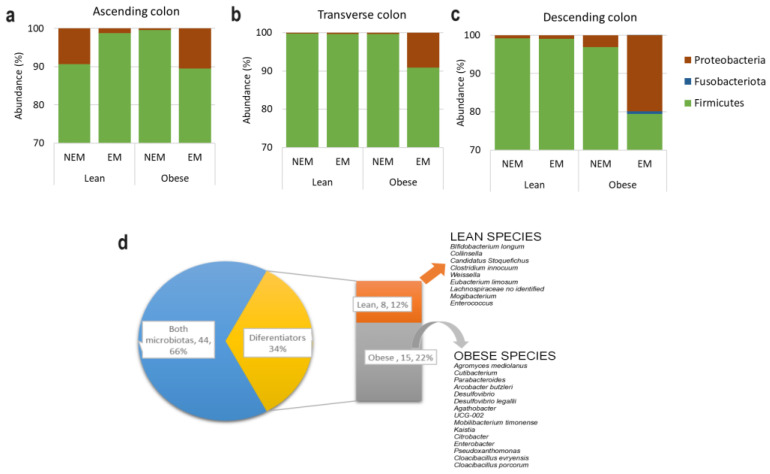
Distribution of the species found in the microbiota of participants who are lean and participants with obesity (**a**–**c**) and common and different species found in each microbiota (**d**).

**Figure 5 foods-13-00587-f005:**
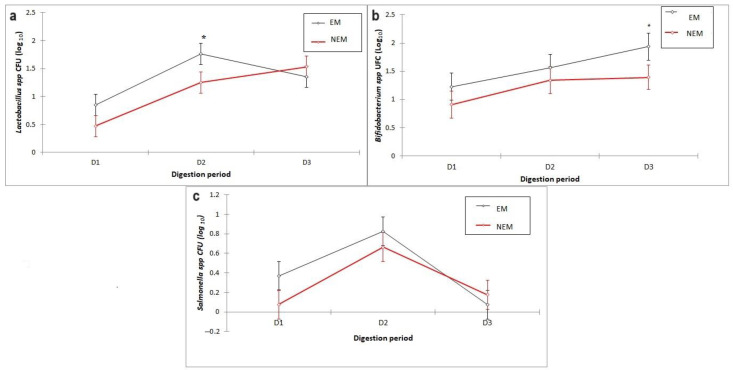
Growth kinetics of (**a**) *Lactobacillus* spp., (**b**) *Bifidobacterium* spp., and (**c**) *Salmonella* spp. via the digestive process + D1 = cycle 1 (day 3), D2 = cycle 2 (day 6), and D3 = cycle 3 (day 9). The bar represents the standard deviation of the samples (*n* = 3), and * means a significant difference between samples *p* < 0.005 (by Tukey’s multiple range test).

**Figure 6 foods-13-00587-f006:**
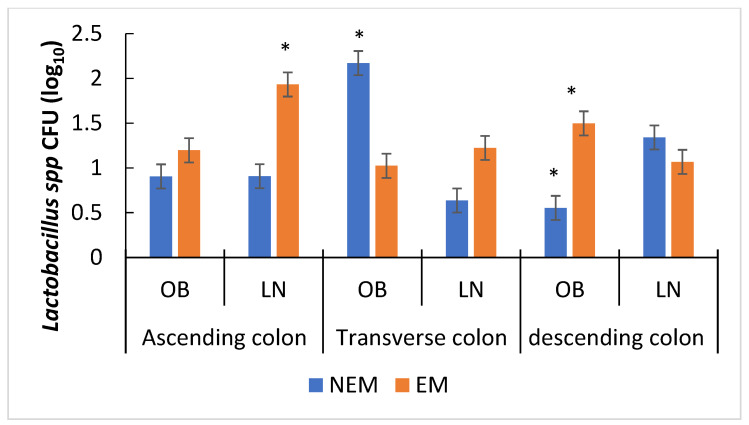
Growth of *Lactobacillus* spp. when administering the different food matrices (NEM = no extract matrix, EM = extract matrix) in the reactors. OB = obese, LN = lean. The column bars represent the sample’s standard deviation (*n* = 3); * means a significant difference between bars of the same group *p* < 0.005 (by Tukey’s multiple range test).

**Figure 7 foods-13-00587-f007:**
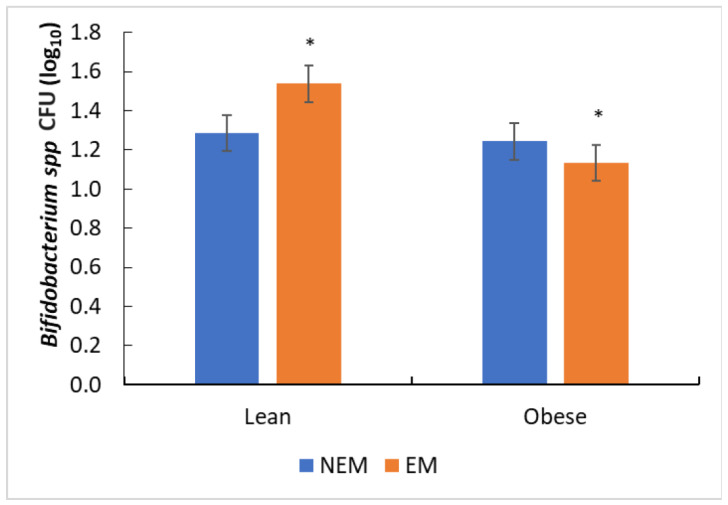
Growth of *Bifidobacterium* spp. after administering food matrices (NEM = no extract matrix, EM = extract matrix). The column bars represent the sample’s standard deviation (*n* = 3), and *** on the columns show significant differences between the samples (by Tukey’s multiple range test).

**Figure 8 foods-13-00587-f008:**
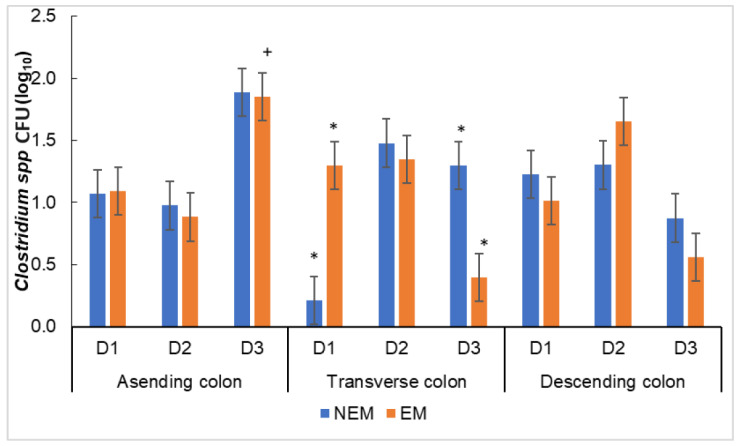
Growth of *Clostridium* spp. when administering the food matrices (NEM = no extract matrix, EM = extract matrix) in the reactors. D1 = day 3, D2 = day 6, and D3 = day 9. The column bars represent the sample’s standard deviation (*n* = 3); the asterisk represents significant differences between the extract matrix and the no extract matrix, and + represents significant differences between the digestive cycles of the same portion of the colon (by Tukey’s multiple range test).

**Figure 9 foods-13-00587-f009:**
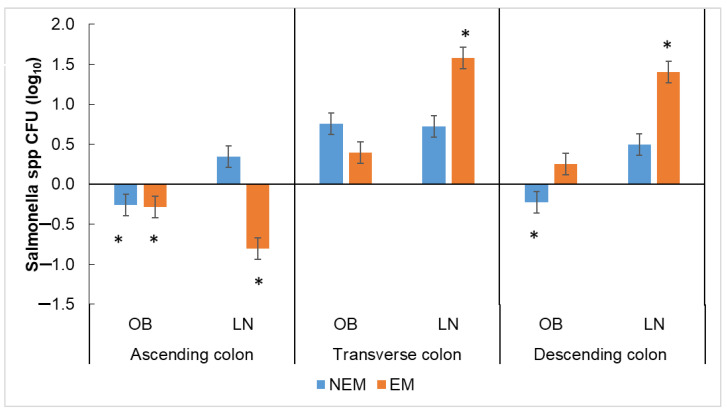
Growth of *Salmonella* spp. in the simulation of different portions of the colon inoculated with the microbiota of people who are lean (LN) and people who have obesity (OB) people after the administration of different matrices (NEM = no extract matrix, EM = extract matrix). The column bars represent the sample’s standard deviation (*n* = 3); the asterisk represents significant differences between the extract matrix and the no extract matrix (by Tukey’s multiple range test).

**Figure 10 foods-13-00587-f010:**
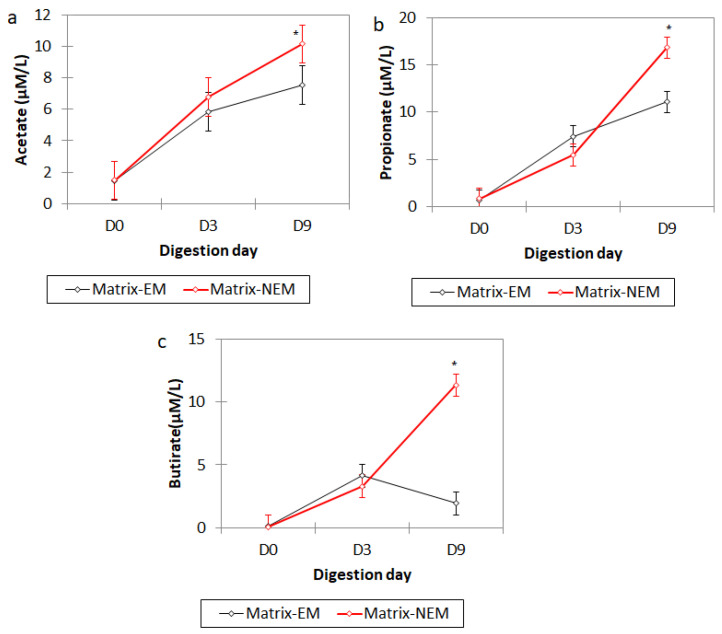
Production of (**a**) acetate, (**b**) propionate, and (**c**) butyrate via the digestion process when administering the different matrices (NEM = no extract matrix, EM = extract matrix, D0 = day 0, D3 = day 3, and D9 = day 9). The column bars represent the sample’s standard deviation (*n* = 3); the asterisk represents significant differences between the extract matrix and the no extract matrix (by Tukey’s multiple range test).

**Figure 11 foods-13-00587-f011:**
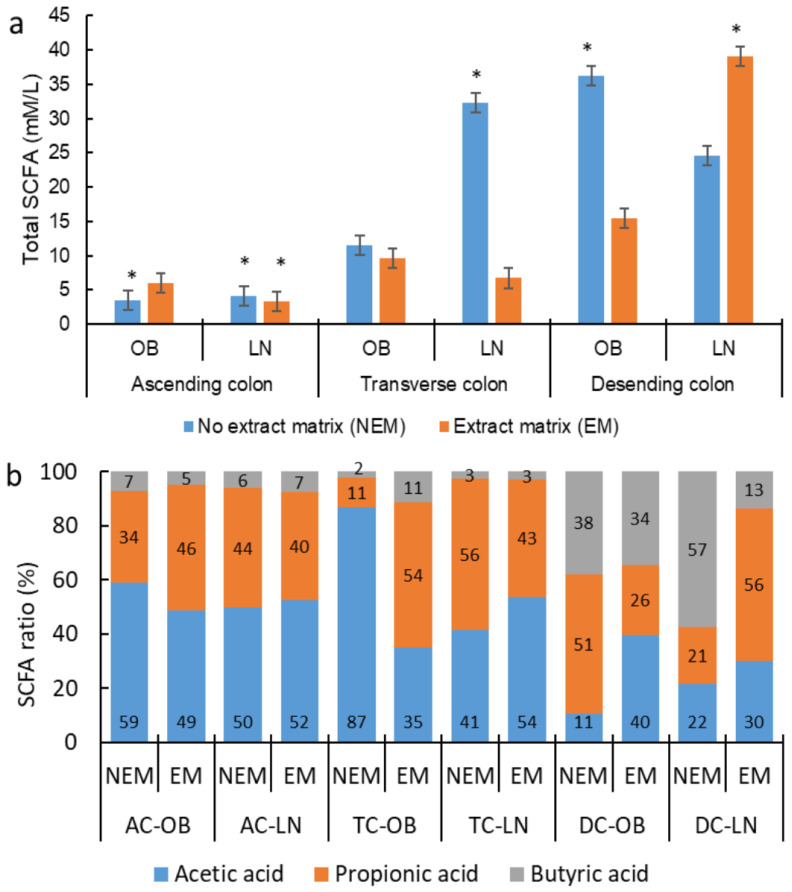
Changes in the concentration of total short-chain fatty acids SCFA (**a**) and change in ration of acetic, propionic and butyric acids (**b**) in the simulation of the different portions of the colon inoculated with the microbiota of lean people (LN) and people who have obesity (OB) people after the administration of different matrices. The column bars represent the sample’s standard deviation (*n* = 3); * means significant differences between samples by Tukey’s multiple range test.

**Table 1 foods-13-00587-t001:** Taxonomic analysis and the distribution of the bacterial families in the microbiota of lean people and people with obesity.

Bacterial Family	Obese Microbiota						Lean Microbiota					
	*AC		**TC		***DC		AC		TC		DC	
	*EM	*NEM	EM	NEM	EM	NEM	EM	NEM	EM	NEM	EM	NEM
*Lachnospiraceae*	X	X	X	X	X	X	X	X	X	X	X	X
*Enterobacteriaceae*	X	X	X	X	X	X						
*Mitochondria*	X	X	X	X	X	X						
*Morganellaceae*	X	X	X	X	X	X						
*Rhizobiaceae*	X	X	X	X	X	X						
*Sphingomonadaceae*	X	X	X	X	X	X						
*Microbacteriaceae*		X	X	X	X	X						
*Xanthobacteraceae*	X	X	X		X	X						
*Beijerinckiaceae*			X	X	X	X						
*Comamonadaceae*	X		X		X	X						
*Xanthomonadaceae*			X		X	X						
*Alcaligenaceae*			X		X							
*Arcobacteraceae*					X	X						
*Fusobacteriaceae*					X	X						
*Kaistiaceae*					X	X						
*Sutterellaceae*					X	X						
*Tannerellaceae*					X	X						
*Synergistaceae*					X							
*Erysipelatoclostridiaceae*							X	X	X	X	X	X
*Erysipelotrichaceae*							X	X	X	X	X	X
*Bifidobacteriaceae*									X	X	X	X
*Coriobacteriaceae*											X	X
*Leuconostocaceae*											X	X
*Acidaminococcaceae*	X	X	X		X	X	X	X	X	X	X	X
*Enterococcaceae*	X	X	X	X	X	X	X	X		X	X	X
*Lactobacillaceae*	X	X	X	X	X	X	X	X		X	X	X
*Eubacteriaceae*			X	X	X	X	X	X	X	X	X	X
*Chloroplast*	X	X	X	X	X	X	X	X		X		
*Veillonellaceae*	X	X	X	X	X	X	X	X				
*Anaerovoracaceae*					X	X	X	X	X	X		
*Oscillospiraceae*	X		X		X	X	X	X				
*Atopobiaceae*					X	X	X				X	X
*Desulfovibrionaceae*					X	X			X		X	X
*Propionibacteriaceae*				X				X				

*AC = ascending colon, **TC = transverse colon, ***DC =descending colon, *EM = extract matrix, *NEM = no extract matrix.

## Data Availability

The original contributions presented in the study are included in the article, further inquiries can be directed to the corresponding author.

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
