# Peer review of "The Effect of Opuntia ficus Mucilage Pectin and Citrus aurantium Extract Added to a Food Matrix on the Gut Microbiota of Lean Humans and Humans with Obesity"

_foods, 2024, doi:10.3390/foods13040587_

Round 1
Reviewer 1 Report
Comments and Suggestions for Authors
Dear authors
Thank you for your excellent manuscript. It is written well ans it's idea is very important. Gut microbiome plays an important in human health. Dysbiosis in gut microbiome contributes in many chronic diseases such as obesity, diabetes, CVD and Alzehimer's disease. So, studying the gut microbiome difference in obese and lean subjects is soo important.
Authors used a complete simulation of the digestion process for studying the difference in gut microbiome of obese and lean subjects and also the impact of extract as perbiotics on the type of bacteria in both studied groups.
The results of the experiment is ideal and logic and discussied well.
Please response to the few changes in the attched PDF file.

Author Response
They have been of great importance in improving the quality of the manuscript. Please, the detail and description of the corrective actions that you kindly find in the attached file. Thanks a lot

Reviewer 2 Report
Comments and Suggestions for Authors
This work aimed to evaluate the effect of a food gel matrix with Opuntias ficus cladodes mucilage pectin and Citrus Aurantium extract on the growth of four beneficial gut bacteria obtained from the fecal microbiota of lean or obese people after digestion in the upper digestive system. This study is great, however requires several changes prior the acceptance for publication:
Abstract:
To add more numbers.
Introduction:
-What is hypothesis of study?
-Introduction is too long.
-Figure 1 is not adequate in introduction.
Methods:
-What is design of study?
-What is sample size calculus?
-What is regsitration of study as clinical trial?
Results:
-What are body composition of patients? %fat mass, lean mass, BMI?
Discussion:
This is speculative. What is discussion regarding to body composition?
Author Response
They have been of great importance in improving the quality of the manuscript. Please, the detail and description of the corrective actions that you kindly find in the attached file. Thanks a lot.

Reviewer 3 Report
Comments and Suggestions for Authors
Comments in attached file.

Quality of English is satisfactory.
Author Response

(The authors gave the same response as above.)

Reviewer 4 Report
Comments and Suggestions for Authors
the authors select a very interesting and actual research topic. the influence of gut microbiota on human health. the research is very well constipated are the obtained results are presented very well. the introduction is very informative with enough information about similar previous research and underlining the importance of the microbiome on human health. the materials and methods are well organized with enough information for the reader to repeat the experiments or do similar research. the results are presented very well and the reader can easily understand the main research focus
the number of figures and tables is adequate and well-analyzed
the conclusion is supported by the results.
However, I have a few doubts which authors should clarify:
first, the authors analyze just short-chain fatty acids I need an explanation as to why the long-chain fatty acids were avoided.
second in general authors said that SCFA was analyzed but without information about the method ( linearity, LOD, LOQ recovery, method accuracy.....)
the aforementioned remark could be applied to polyphenols also.
what about the matrix effect in TPC determination I understand that this was an artificially made intestinal model but it should be taken into account at least in discussion.
Comments on the Quality of English Languagethe paper is easy to read and understand hence the English style and grammar are satisfactory.
Author Response

(The authors gave the same response as above.)
